# Physics Guided Neural Networks for Spatio-temporal Super-resolution of Turbulent Flows

**Tianshu Bao**[*1]     **Shengyu Chen**[*2]     **Taylor T. Johnson**[1]     **Peyman Givi**[3]     **Shervin Sammak**[3]     **Xiaowei Jia**[2]

[1] Department of Computer Science, Vanderbilt University, Nashville, Tennessee, USA
[2] Department of Computer Science, University of Pittsburgh, Pittsburgh, Pennsylvania, USA
[3] Department of Mechanical Engineering and Materials Science, University of Pittsburgh, Pittsburgh, Pennsylvania, USA

## Abstract

Direct numerical simulation (DNS) of turbulent flows is computationally expensive and is not practical for simulating flows at high Reynolds numbers. Low-resolution large eddy simulation (LES) is a pragmatic alternative, but its success depends on modeling of the small scale flow dynamics. Reconstructing DNS from low-resolution LES is critical for many scientific and engineering disciplines, but it poses many challenges to existing super-resolution methods due to the complexity of turbulent flows and computational cost of generating frequent LES data. In this work, we propose a physics-guided neural network for reconstructing frequent DNS from sparse LES data by enhancing its spatial resolution and temporal frequency. Our proposed method consists of a partial differential equation (PDE)-based recurrent unit for capturing underlying temporal processes and a physics-guided super-resolution model that incorporates additional physical constraints. We demonstrate the effectiveness of both components in reconstructing the data generated by simulating the Taylor-Green Vortex sparse LES data. Moreover, we show that the proposed recurrent unit can preserve the physical characteristics of turbulent flows by leveraging the physical relationships in the Navier-Stokes equation.

## 1 INTRODUCTION

Understanding turbulence is the key to our comprehension of many natural and technological processes in engineering, science, medicine and many other disciplines. Direct numerical simulation (DNS) of the Navier-Stokes equations is widely regarded as the methodology with the highest fidelity

in capturing the dynamics of turbulent flows (Givi [1994]). DNS is essentially a brute force computational methodology to provide solution of the unsteady governing equations of fluid flow at all temporal and spatial scales. Such simulations can be very expensive at high Reynolds numbers. A practical alternative, the large eddy simulation (LES) concentrates on the larger scale eddies and models the effects of the subgrid-scale transport. By this filtering, LES can be conducted on coarser grids as compared to those required by DNS. The penalty is that LES-generated data are, generally, of lower fidelity as compared to DNS (Nouri et al. [2017]).

Machine learning, especially super-resolution (SR) methods (Park et al. [2003]), has already shown tremendous success in reconstructing high-resolution data in a variety of commercial applications. The power of these models comes mainly from the use of convolutional network layers (Albawi et al. [2017]), which can extract the spatial texture features and transform them through complex non-linear mappings to recover high-resolution data. From the earliest end-to-end convolution-based SR model (Dong et al. [2014]), many investigators have added skip-connections in SR models (Zhang et al. [2018a], Van Duong et al. [2021], Dai et al. [2019], Zhang et al. [2018b], Ahn et al. [2018], Tai et al. [2017]) to bypass redundant low-resolution information and promote the stability of optimizing deep networks. Moreover, advances in adversarial learning allow preservation of high-level features extracted from target high-resolution images through a separate discriminator network (Ledig et al. [2017], Chen et al. [2018], Wang et al. [2018a,b], Karras et al. [2017], Upadhyay and Awate [2019], Cheng et al. [2021], Zhang et al. [2019]). Given their success in computer vision, researchers begin to apply SR methods to reconstruct turbulence data (Fukami et al. [2019], Obiols-Sales et al. [2021], Deng et al. [2019], Stengel et al. [2020], Venkatesh et al. [2021], Xie et al. [2018], Fukami et al. [2020], Liu et al. [2020], Chen et al. [2021]).

However, existing SR methods face several challenges when applied for reconstructing turbulent flows. Such flows involve multiple physical variables and often exhibit complex

---

*These authors contributed equally to this work.

*Accepted for the 38th Conference on Uncertainty in Artificial Intelligence (UAI 2022).*

dynamic patterns, i.e., multiple physical variables evolve and interact at different scales. In the absence of underlying physical processes, pure data-driven SR models require a large number of training samples to capture the correct physics. Due to the substantial computational cost in simulating turbulent flows, high-fidelity DNS data are rarely available, and even the generation of high-quality LES at a lower resolution can be expensive. Hence, low-resolution LES data cannot be frequently generated for a large variety of scenarios. When trained with limited data at discrete time steps (i.e., when both LES and DNS are available), these models can have degraded performances because they may learn spurious patterns between sparse observations, and such patterns are often not generalizable.

In this work, we propose a new physics-guided neural network framework for spatial and temporal super-resolution. The idea is to leverage underlying physical relationships to guide the learning of generalizable spatial and temporal patterns in the reconstruction process. In particular, our framework consists of two components, physics-guided recurrent unit (PRU) and physics-guided super resolution model (PGSR). The PRU structure is designed based on the underlying partial differential equation (PDE), and is responsible for capturing the temporal dynamics of turbulent flows from sparse data. The PGSR model incorporates additional physical constraints to improve the reconstruction from the available LES data. Our evaluation of the Taylor-Green Vortex data (Brachet et al. [1984]) has demonstrated the superiority of PRU and PGSR in modeling the turbulent flows. At the same time, we also verify that the proposed method can preserve the physical properties of turbulent flows.

Our contributions can be summarized as:

- We propose innovative physics-guided PRU and PGSR architectures to capture the temporal and spatial patterns of the turbulent flows, respectively.
- We design a unified neural network framework combining PGSR and PRU to effectively simulate and reconstruct high-resolution frequent turbulent flows.
- We evaluate our model in a series of experiments. The experimental results demonstrate that our approaches have significant superiority compared with existing methods in both DNS simulation from historical data and DNS reconstruction from sparse LES data.

## 2 PROBLEM DEFINITION

Our objective is to reconstruct frequent high-resolution flow data from low-resolution and sparse LES data. In particular, we consider a general three-dimensional vortex flow over space and time $Q(x, y, z, t)$, where $(x, y, z)$ denotes the spatial coordinates, $t$ represents the time step (in seconds), and $Q(x, y, z, t)$ consists of multiple variables that

describe turbulent transport, such as the velocity along with different directions and the thermodynamic pressure. We represent low-resolution LES data as $Q^{LR}(x, y, z, t)$, which are available at sparse time steps, e.g., starting from a time step $t_0$, the LES is generated with a time interval of $d$ at $\{t_0, t_0 + d, t_0 + 2d, ...\}$. The flow variables in $Q(x, y, z, t)$ also follow the Navier-Stokes equation, which governs the transport of these variables in space $(x, y, z)$ and time $(t)$. Boundary conditions are specified near the boundary of the domain to describe the interaction of the flow with the external environment. More details about the flow dataset will be provided in Section 4.1.

## 3 METHOD

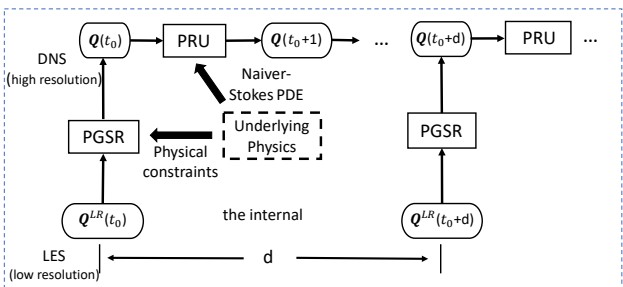

Figure 1: The proposed physics-guided neural networks framework combining PRU and PGSR for reconstructing turbulent flows $Q$.

Our proposed framework consists of two structural components, PGSR and PRU, which are illustrated in Fig. 1. Starting from an initial time step $t_0$, the proposed method will follow a two-step process: (i) the PGSR model is used to reconstruct high-resolution $Q(x, y, z, t)$ when low-resolution LES data are available. (ii) Then PRU is used to estimate $Q(x, y, z, t + 1)$ from $Q(x, y, z, t)$ until the next LES sample is available. In the following, we will describe these two components: PRU and PGSR.

### 3.1 PHYSICS-GUIDED RECURRENT UNIT (PRU)

Physical variables $Q$ in turbulent flows interact with each other and evolve at different speeds for different locations. Temporal neural network models, e.g., long-short term memory (LSTM) (Hochreiter and Schmidhuber [1997]), have sophisticated structures and thus heavily rely on large representative training data that are sampled at the high temporal frequency to capture the underlying continuous patterns over time. Given sparse and limited LES data, we come up the PRU structure as a more accurate and reliable way to predict the future flow variables by leveraging the continuous physical relationship described by the underlying PDE. This helps bridge the gap between discrete data samples and continuous flow dynamics. The proposed PRU structure

is inspired by our previous work on combining machine learning and physical equations Jia et al. [2019], Bao et al. [2021], Jia et al. [2021], Willard et al. [2021]. The PRU structure is also generally applicable to many dynamical systems with governing PDEs.

Most PDEs can be represented in the form of $\boldsymbol{Q}_t = \boldsymbol{f}(t, \boldsymbol{Q}; \theta)$, where $\boldsymbol{Q}_t$ is the temporal derivative of $\boldsymbol{Q}$, and $\boldsymbol{f}(t, \boldsymbol{Q}; \theta)$ is a non-linear function (parameterized by coefficient $\theta$) that summarizes the current value of $\boldsymbol{Q}$ and its spatial context. For example, the incompressible Navier-Stokes equation for the velocity field can be expressed as:

$$\boldsymbol{f}(\boldsymbol{Q}) = \frac{-1}{\rho}\nabla p + \nu\Delta\boldsymbol{Q} - (\boldsymbol{Q}.\nabla)\boldsymbol{Q}, \qquad (1)$$

where $\rho$, $p$, and $\nu$ denote the fluid density, the thermodynamic pressure, and the viscosity, respectively. Since the function $\boldsymbol{f}(\boldsymbol{Q})$ in the Navier-Stokes equation is independent of time $t$, we omit the independent variable $t$ in the function $\boldsymbol{f}(\cdot)$. Here $p$ is treated as a known variable, and $\theta = \{\rho, \nu\}$.

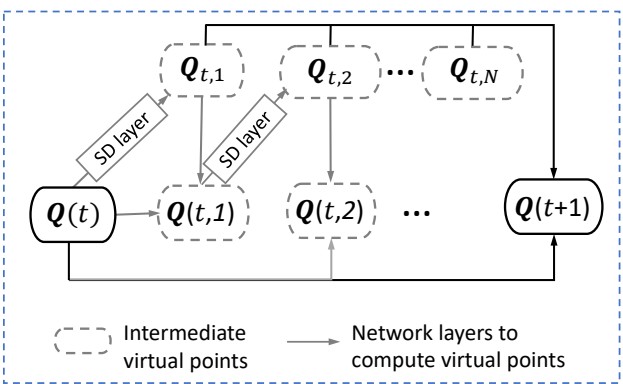

Intermediate virtual points — Network layers to compute virtual points

Figure 2: Diagram for the physical recurrent unit, which iteratively estimates the temporal derivative and the intermediate state variable.

The PRU structure is inspired by the classical numerical Runge–Kutta (RK) methods (Butcher [2007]), which have been used in temporal discretization for the approximate solutions of differential equations. As shown in Fig. 2, the central idea of PRU is to interpolate virtual intermediate variables and create smaller intervals between two time steps which facilitate refining the gradient of flow variables over time. Starting from a time step $t$, PRU estimates $N-1$ intermediate state variables $\boldsymbol{Q}(t, 1)$, ..., $\boldsymbol{Q}(t, N-1)$ and $N$ intermediate temporal derivatives $\boldsymbol{Q}_{t,1}$, ..., $\boldsymbol{Q}_{t,N}$ before reaching the next step $t+1$.

In particular, PRU interpolates intermediate state variables by iteratively following a two-step process: for $n$ from 1 to $N$, (i) PRU first estimates the temporal derivative $\boldsymbol{Q}_{t,n} = \boldsymbol{f}(\boldsymbol{Q}(t, n-1))$ at the previous intermediate flow state $\boldsymbol{Q}(t, n-1)$, and $\boldsymbol{Q}(t, 0) = \boldsymbol{Q}(t)$. We will discuss more details about how to compute the function $\boldsymbol{f}(\cdot)$ later. (ii) Then PRU computes the next intermediate state variable

$\boldsymbol{Q}(t, n)$ by moving the flow data $\boldsymbol{Q}(t)$ along the direction of obtained temporal derivatives. In our tests, we follow the most popular $4^{th}$ order RK method for computing the three intermediate state variables, as follows:

$$\begin{aligned} \boldsymbol{Q}(t, 1) &= \boldsymbol{Q}(t) + \Delta t\frac{\boldsymbol{Q}_{t,1}}{2}, \\ \boldsymbol{Q}(t, 2) &= \boldsymbol{Q}(t) + \Delta t\frac{\boldsymbol{Q}_{t,2}}{2}, \qquad (2) \\ \boldsymbol{Q}(t, 3) &= \boldsymbol{Q}(t) + \Delta t\boldsymbol{Q}_{t,3}, \end{aligned}$$

The temporal derivative $\boldsymbol{Q}_{t,4}$ is then computed from the last intermediate point, as $\boldsymbol{f}(\boldsymbol{Q}(t, 3))$. The $4^{th}$ order RK method has the total accumulated error of $O(\Delta t^4)$, where $\Delta t$ represents the time interval between consecutive time steps.

Finally, PRU combines all the intermediate temporal derivatives as a composite gradient to predict the flow variables at the next time step $\boldsymbol{Q}(t+1)$, as follows:

$$\text{PRU}(\hat{\boldsymbol{Q}}(t+1)|\boldsymbol{Q}(t)) = \boldsymbol{Q}(t) + \sum_{n=1}^{N} w_n\boldsymbol{Q}_{t,n}, \quad (3)$$

where $\{w_n\}_{n=1}^{N}$ are the trainable model parameters. Given a series of high-fidelity DNS training data of $T$ time steps, the PRU structure can be trained by minimizing the mean squared error (MSE) between the predicted flow variables and true DNS values, as $\sum_t ||\text{PRU}(\hat{\boldsymbol{Q}}(t+1)|\boldsymbol{Q}(t)) - \boldsymbol{Q}(t+1)||^2/T$.

In the following, we will describe two major issues in computing the function $\boldsymbol{f}(\cdot)$: (i) estimating spatial derivatives in the function $\boldsymbol{f}(\cdot)$, and (ii) preserving boundary conditions. We will also investigate the stability of this method for long-term prediction with a simple case study.

### 3.1.1 Spatial derivative approximation

The proposed PRU evaluates the function $\boldsymbol{f}(\cdot)$ explicitly for estimating the temporal derivatives of intermediate state variables. In many general PDEs (e.g., the Navier-Stokes equation), $\boldsymbol{f}(\boldsymbol{Q})$ contains spatial derivatives of $\boldsymbol{Q}$. One popular approach for evaluating the spatial derivatives is through the finite difference methods (FDMs), which approximate variable derivatives of a function on predefined mesh points by solving algebraic equations containing finite differences and values from nearby points. For example, the first and second order spatial derivatives along the $x$ dimension in Eq. 1 (represented as $\boldsymbol{Q}_x$ and $\boldsymbol{Q}_{xx}$) can be estimated by the

FDMs as follows:

$$
\begin{aligned}
& \boldsymbol{Q}_x(x_i, y_j, z_k, t_n) \\
& \approx \frac{\boldsymbol{Q}(x_{i+1}, y_j, z_k, t_n) - \boldsymbol{Q}(x_{i-1}, y_j, z_k, t_n)}{2\Delta x}, \\
& \boldsymbol{Q}_{xx}(x_i, y_j, z_k, t_n) \\
& \approx [\boldsymbol{Q}(x_{i+1}, y_j, z_k, t_n) - 2\boldsymbol{Q}(x_i, y_j, z_k, t_n) \\
& + \boldsymbol{Q}(x_{i-1}, y_j, z_k, t_n)]/(\Delta x)^2.
\end{aligned}
\tag{4}
$$

The approximation used in FDMs results in an error compared to the exact solution, which can be estimated through Taylor expansions. Instead of using FDMs for every mesh point, we propose to build a spatial difference (SD) layer using convolutional neural network (CNN) layers. The CNN layers have the expressive power to capture the relationships defined in FDMs (Eq. 4) while also being more flexible in learning other non-linear relationships from data.

### 3.1.2 Boundary Condition and Augmentation

Boundary conditions are critical in turbulent flow simulation as they describe how the turbulent flows interact with the external environment. Here we consider the periodic boundary condition in our flow data. It is defined in a specified periodic domain indicating that it repeats its own values in all directions. The formal definition of a cubic periodic boundary condition is given below:

$$
\begin{aligned}
\boldsymbol{Q}(L_x, y, z, t) &= \boldsymbol{Q}(R_x, y, z, t), \\
\boldsymbol{Q}(x, L_y, z, t) &= \boldsymbol{Q}(x, R_y, z, t), \\
\boldsymbol{Q}(x, y, R_z, t) &= \boldsymbol{Q}(x, y, R_z, t),
\end{aligned}
\tag{5}
$$

where $L_x, L_y, L_z$ are the three left boundaries with respect with $x, y, z$ coordinates and $R_x, R_y, R_z$ are the three right boundaries with respect with $x, y, z$ coordinates. Standard padding strategies for CNN (e.g., same padding) do not satisfy the periodic value requirement. In order to handle this issue, we make a data augmentation for each of the 6 faces (of the 3D cubic data) with an additional 2 layers of data during the training stage and adopt a $5 \times 5$ CNN filter size. The augmented locations will be removed from reconstructed data.

### 3.1.3 Stability

The classical $4^{th}$ order RK suffers from the stability issue if the step size is not properly chosen. Consider a simple scalar example $Q_t = \lambda Q$. The $4^{th}$ order RK for this equation can be written as

$$
\begin{aligned}
& Q((n+1)\Delta t) \\
& \approx (1 + \lambda \Delta t + \frac{\lambda \Delta t^2}{2} + \frac{\lambda \Delta t^3}{6} + \frac{\lambda \Delta t^4}{24}) Q(n\Delta t).
\end{aligned}
\tag{6}
$$

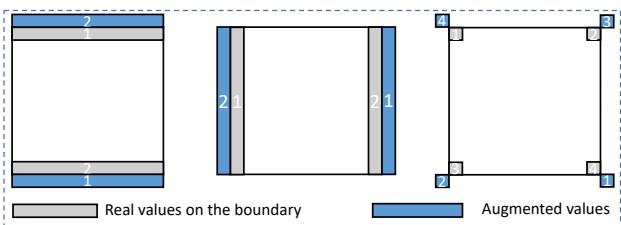

Figure 3: Illustration of data augmentation on a 2-D example. The left diagram represents the up and low boundary augmentation. The middle diagram represents the left and right boundary augmentation. And the right diagram represents the corner boundary augmentation. Rectangles carrying identical numbers have the same value.

Let's denote $R(\Delta t) = 1 + \Delta t + \frac{\Delta t^2}{2} + \frac{\Delta t^3}{6} + \frac{\Delta t^4}{24}$, and we have $Q((n+1)\Delta t) = R(\Delta t)Q(n\Delta t)$. The analytical solution is $Q((n+1)\Delta t) = \exp(\lambda \Delta t)Q(n\Delta t)$, and thus the accumulated error is

$$
err_{n+1} = (\exp(\lambda \Delta t) - R(\Delta t))err_n.
\tag{7}
$$

This indicates that $err_{n+1} = O(\Delta t^5)err_n$ according to Taylor expansion. When the interval $d$ of LES data is large, the accumulated error may get amplified at every time step and then lead to an explosion. Additional complexity arises when $f$ consists of multiple evaluations of spatial derivatives. This requires the access to LES data at a reasonably frequent time interval to avoid significantly large reconstruction errors.

---

**Algorithm 1** The flow of the proposed PRU.

---

Create and initialize $5 \times 5$ filters for $1^{st}$ and $2^{nd}$ order spatial derivatives
**for** $epoch$ = 1 : number of training iterations **do**
  **for** $t$ = 1 : number of time steps **do**
    Make data augmentation for $\boldsymbol{Q}(t)$ (Section 3.1.2).
    Calculate $\boldsymbol{Q}_{t,1}, \boldsymbol{Q}_{t,2}, \boldsymbol{Q}_{t,3}, \boldsymbol{Q}_{t,4}$ following Eq. 2 and evaluate $\boldsymbol{f}$ accordingly.
    Calculate $\hat{\boldsymbol{Q}}(t+1)$ following Eq. 3 and remove augmented data over boundaries.
    Use the predicted $\hat{\boldsymbol{Q}}(t+1)$ as the input flow data for time $t+1$.
  **end for**
  Update trainable filters and weights.
**end for**

---

## 3.2 PHYSICS GUIDED SUPER RESOLUTION (PGSR)

The PGSR model aims to incorporate additional physical constraints to regularize the standard super-resolution model. In particular, we consider two important physical constraints,

the divergence-free property for the incompressible flow and the zero-mean property for the Taylor-Green Vortex (Brachet et al. [1984]).

First, the incompressible flow follows the divergence-free property in the velocity field. Thus, we can represent the inherent physical relationship of the velocity field as:

$$\nabla \cdot \mathbf{V} = \frac{\partial u}{\partial x} + \frac{\partial v}{\partial y} + \frac{\partial w}{\partial z} = 0, \qquad (8)$$

where we represent the velocity vector $\mathbf{V}(\mathbf{x}, t)$ along 3-D dimensions ($\mathbf{x} \equiv x, y, z$) by $u$, $v$, and $w$, respectively. Then we use a second-order central finite difference approximation to estimate the partial derivatives and employ this divergent free property as a physical loss in the training process, as follows:

$$\mathcal{L}_{\text{Phy}} = \sum_{(x,y,z)} \left[ \nabla \cdot \hat{\mathbf{V}}(\mathbf{x}, t) \right]^2 / M, \qquad (9)$$

where $M$ is the number of spatial locations in the high-resolution data, and $\hat{\mathbf{V}}$ represents the reconstructed velocity field at high resolution. Such physical constraint can help reduce the search space for model parameters such that the reconstructed high-resolution data follow the divergence-free property which is enforced in incompressible flows.

Second, to preserve the zero-mean property of the in a compressible flow, we also implement an extra network layer by reducing the mean value of reconstructed flows in the generative process $\hat{Q}_0 = g(\hat{Q})$. We do not include the zero-mean constraint directly in the loss function because the obtained model cannot preserve the zero-mean property for the long-term testing phase. On the other hand, direct MSE minimization using $\hat{Q}_0$ as output leads to an unstable training process because the original output $\hat{Q}$ can have arbitrarily large values. Hence, we iteratively train the PGSR model to reduce (i) the gap between $\hat{Q}$ and the true DNS, (ii) the gap between $\hat{Q}$ and its resulted $\hat{Q}_0$, and finally use $\hat{Q}_0$ as the output.

Additionally, we also introduce a degradation process to enforce the consistency between the reconstructed data and the input LES data, similar to Chen et al. [2021]. We create the PGSR model based on the popular SR model SR-GAN (Ledig et al. [2017]). The methods we used to include physical constraints can easily be applied into enhance other SR models as well.

# 4  EXPERIMENT

In this section, we evaluate the performance of our method on a Taylor-Green vortex (TGV) (Brachet et al. [1984]) dataset and compare the results with existing well-used methods. We first introduce the dataset used in our tests, and discuss the experimental design and evaluation targets. Then we will provide experimental results and our analysis.

## 4.1  DATASET

We consider a variant of the Taylor-Green vortex (TGV). This is a three-dimensional incompressible flow and is simulated within a box with periodic boundary conditions. The TGV provides a suitable setting for our demonstration as it exhibits several salient features of turbulent transport. In this flow, the original vortex collapses into turbulent worm-like structures which become progressively more turbulent until viscosity eventually dissipates the large scale vortical structures. We compare our proposed method against several existing super-resolution algorithms to reconstruct the DNS data of TGV.

The TGV is produced by a solution of the constant density Navier-Stokes equation:

$$\frac{\partial \mathbf{V}}{\partial t} + (\mathbf{V}.\nabla)\mathbf{V} = \frac{-1}{\rho}\nabla p + \nu \Delta \mathbf{V}. \qquad (10)$$

The evolution of the TGV includes enhancement of vorticity stretching and the consequent production of small-scale eddies. Initially, large vortices are placed in a cubic periodic domain of $[-\pi, \pi]$ (in all three-directions), with initial conditions:

$$\begin{align}
u(x, y, z, 0) &= \sin(x)\cos(y)\cos(z) & (11) \\
v(x, y, z, t) &= -\cos(x)\sin(y)\cos(z) & (12) \\
w(x, y, z, t) &= 0. & (13)
\end{align}$$

Then the value of the Reynolds number is set to $Re = 1600$. We have LES and DNS results of TGV at several times steps. For each time step, we consider the three components of the velocity along the $x$, $y$, and $z$ axis, denoted by $u$, $v$, and $w$, respectively. Our objective is to reconstruct the DNS results of the velocity field $(u, v, w)$ using LES data. In particular, $Q^{LR}$ represents the LES values of the velocity field while the target $Q$ represents the high-fidelity DNS of the velocity field. Here both LES and DNS data are generated along 65 grid points along the $z$ axis under equal intervals. The LES and DNS are conducted on 32-by-32 and 128-by-128 grid points, respectively, along the $xy$ directions. Hence, the DNS data is of 16 times higher resolution compared to LES data.

## 4.2  EXPERIMENTAL DESIGN

We train the proposed method using the TGV data from a consecutive 20-seconds period (with 20 time steps) and then apply the trained model to the next 50 seconds' testing data and measure the performance. [1] We evaluate the performance of DNS prediction using two different evaluation

---

[1]Code for the experiment is available at drive.google.com/drive/folders/11PTaEjsBkgd6PAAYm WH_KDzTg90IvrJn?usp=sharing

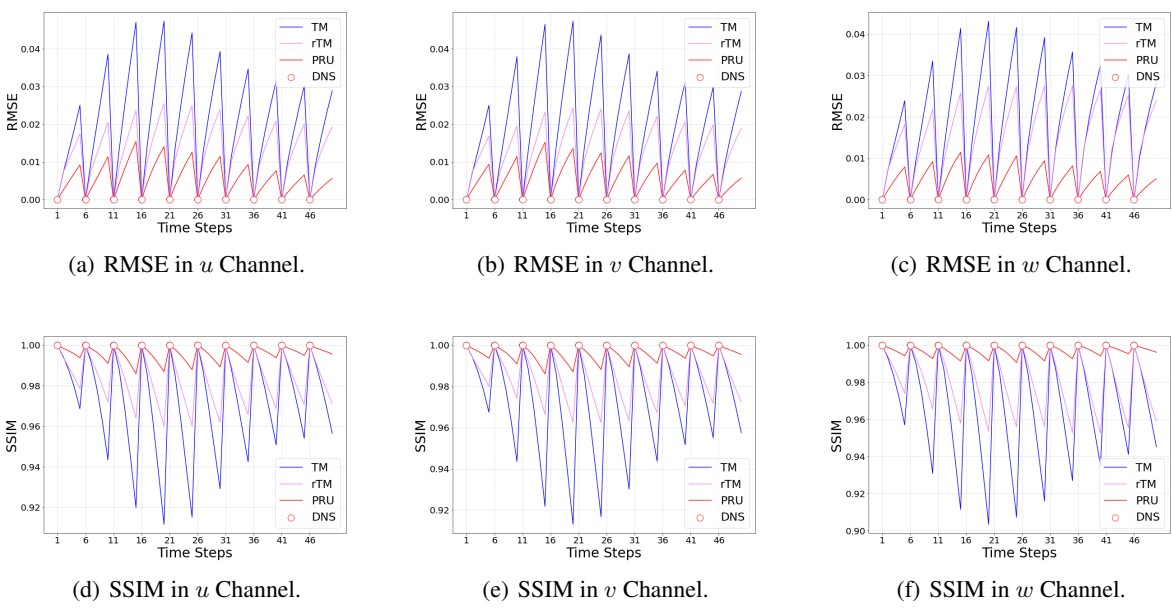

(a) RMSE in $u$ Channel.  (b) RMSE in $v$ Channel.  (c) RMSE in $w$ Channel.

(d) SSIM in $u$ Channel.  (e) SSIM in $v$ Channel.  (f) SSIM in $w$ Channel.

Figure 4: Change of RMSE/SSIM values produced by different DNS prediction models from the 1st to 50th time steps in a testing period with true DNS data for 5 time steps. (a)-(c) show the changes of RMSE values, and (d)-(f) show the changes of SSIM values for $(u,v,w)$ three different channels.

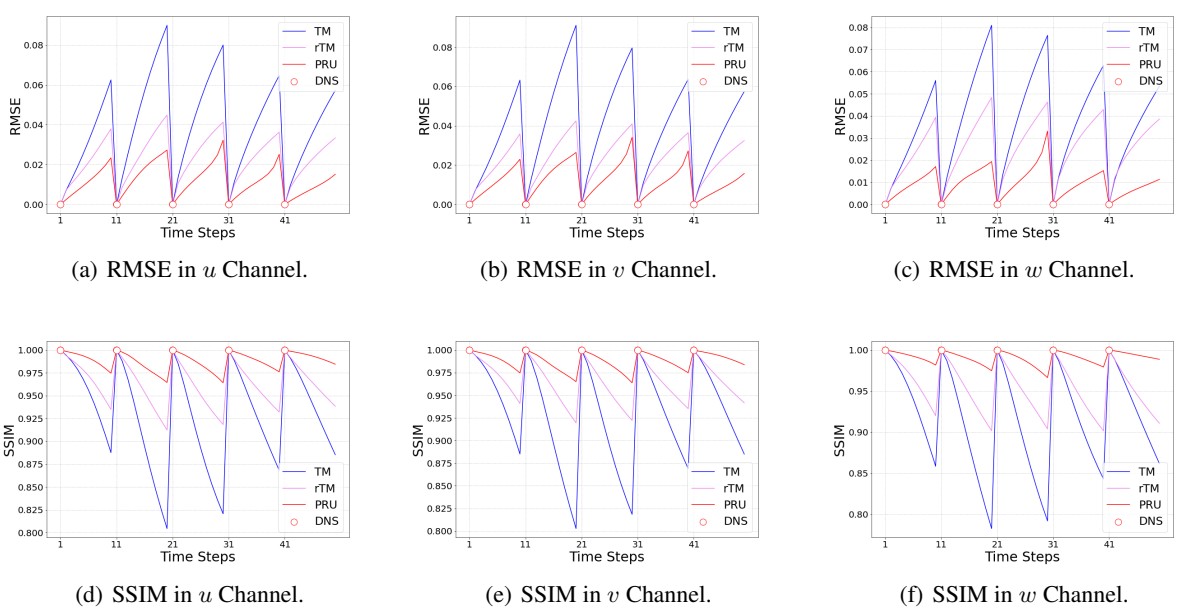

(a) RMSE in $u$ Channel.  (b) RMSE in $v$ Channel.  (c) RMSE in $w$ Channel.

(d) SSIM in $u$ Channel.  (e) SSIM in $v$ Channel.  (f) SSIM in $w$ Channel.

Figure 5: Change of RMSE/SSIM values produced by different DNS prediction models from the 1st to 50th time steps in a testing period with true DNS data for 10 time steps. (a)-(c) show the changes of RMSE values, and (d)-(f) show the changes of SSIM values for $(u,v,w)$ three different channels.

metrics, root mean squared error (RMSE) and structural similarity index measure (SSIM) (Wang et al. [2004]). We use RMSE to measure the difference (error) between reconstructed data and target DNS data. The lower value of RMSE indicates better reconstruction performance at the pixel level. SSIM is used to appraise the structural similarity between reconstructed data and target DNS on three aspects: luminance, contrast, and overall structure.

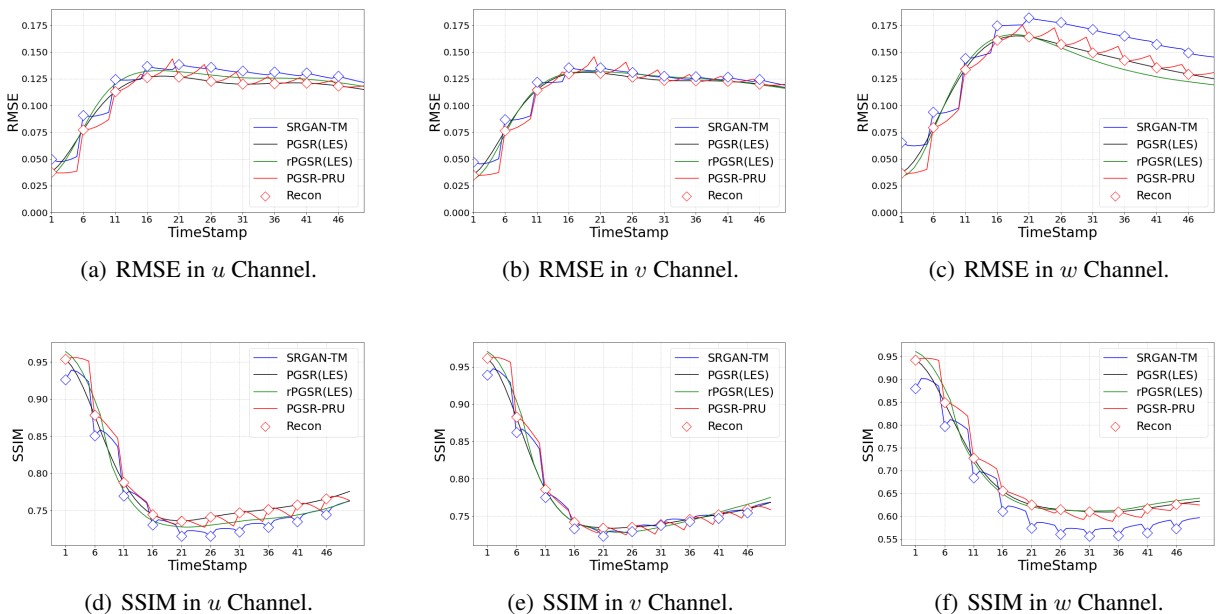

| | | |
|:---:|:---:|:---:|
| (a) RMSE in $u$ Channel. | (b) RMSE in $v$ Channel. | (c) RMSE in $w$ Channel. |
| (d) SSIM in $u$ Channel. | (e) SSIM in $v$ Channel. | (f) SSIM in $w$ Channel. |

Figure 6: Change of RMSE/SSIM for different models over time using sparse LES data with the interval of 5 time steps. (a)-(c) show the changes of RMSE values, and (d)-(f) show the changes of SSIM values for the three different channels $(u,v,w)$.

Table 1: The prediction performance of reconstructing DNS data measured in terms of RMSE and SSIM. The performance is measured on $(u, v, w)$ channels with DNS interval $d$ as 5 or 10. The upper half is the average results using the LES interval of $d = 5$, the bottom half is the average results using the LES interval of $d = 10$.

| Method | RMSE | SSIM |
|:---:|:---:|:---:|
| TM | (0.019,0.019,0.019) | (0.972,0.973,0.967) |
| rTM | (0.013,0.012,0.015) | (0.983,0.984,0.980) |
| PRU | (0.005,0.005,0.005) | (0.996,0.996,0.997) |
| TM | (0.038,0.038,0.036) | (0.930,0.930,0.917) |
| rTM | (0.022,0.022,0.025) | (0.964,0.966,0.954) |
| PRU | (0.012,0.012,0.009) | (0.988,0.988,0.991) |

Our evaluations aim to answer several questions as listed below:

**E1**: *Whether PRU alone can effectively predict the next high-resolution DNS using the previous DNS data?* We compare PRU with two pure data-driven baseline models, transition model (TM) and recurrent transition model (rTM). The TM method predicts the flow variables $\boldsymbol{Q}(t+1)$ at next step using an UNet-style encoder-decoder convolutional structure from the flow variables $\boldsymbol{Q}(t)$ at the previous time. The rTM method further extends TM with a recurrent layer.

**E2**: *Whether the predictions made by PRU can preserve physical properties of DNS?* Besides RMSE and SSIM, we also measure the turbulent kinetic energy of the predicted flows and compare it with that of the true DNS.

**E3**: *How is the reconstruction performance combining PRU and PGSR using sparse low-resolution LES data?* We combine PRU and PGSR (PGSR-PRU) for reconstructing DNS from sparse LES samples. Since we have already compared PRU with other temporal transition models in **E1**, here we compare to a baseline SRGAN-TM, which uses our base SR model SRGAN for reconstructing DNS from LES and use TM to predict DNS when LES is not available. We also compare to another two baselines PGSR (LES) and its extension rPGSR (LES). The rPGSR(LES) method has another recurrent layer over time. Different from PGSR-PRU and SRGAN-TM, these two methods apply the SR model using LES data at all the time steps, thus can be considered as the upper bound for this test. Our goal is to verify that PGSR-PRU can produce comparable performance with PGSR (LES) and rPGSR (LES) while outperforming other baselines.

**E4**: *How is the reconstruction performance of PGSR compared to other SR methods?* We compare PGSR with two well-used SR methods: RCAN (Zhang et al. [2018a]) and SRGAN (Ledig et al. [2017]). We also compare it with DCS/MS (Fukami et al. [2019]), which is a popular SR approach for turbulent flows reconstruction. Additionally, we compare to a variant of PGSR, termed PGSR-D, which only adds the degradation loss to the SRGAN model without using any physical constraints.

Table 2: Reconstruction performance on $(u, v, w)$ using LES channels by RMSE and SSIM. SRGAN-TM and PGSR-PRU (proposed) are evaluated using sparse LES data with the interval of 5 steps, the upper half is the average results of a total of 50 time steps, the bottom half is the average results of the first 15 time steps.

| Method | RMSE | SSIM |
|---|---|---|
| PGSR(LES) | (0.112,0.114,0.133) | (0.771,0.774,0.667) |
| rPGSR(LES) | (0.114,0.114,0.129) | (0.772,0.773,0.669) |
| SRGAN-TM | (0.118,0.115,0.147) | (0.769,0.767,0.647) |
| PGSR-PRU | (0.111,0.113,0.128) | (0.782,0.781,0.681) |
| PGSR(LES) | (0.088,0.088,0.101) | (0.846,0.849,0.801) |
| rPGSR(LES) | (0.091,0.086,0.099) | (0.848,0.855,0.811) |
| SRGAN-TM | (0.091,0.088,0.105) | (0.848,0.849,0.794) |
| PGSR-PRU | (0.081,0.081,0.091) | (0.864,0.866,0.833) |

## 4.3 RESULTS

### 4.3.1 DNS generation using PRU

Here we assume that we have true DNS data with an interval of $d$ time steps ($d = 5$ or $10$) and we implement PRU and other baselines to predict DNS for the missing time steps. We summarize the performance of PRU and baselines in Table 1 and show their performance change on each channel over time in Figs. 4 and 5. For both cases (with the true DNS interval $d$ sets to a larger value 10 or a smaller value 5), PRU produces better performance than baselines over all the time steps. It confirms the effectiveness of PRU in the long-term prediction of DNS from historical flow data (**E1**).

Besides, we compute the kinetic energy of the flow data predicted by PRU and baselines and measure the gap with the kinetic energy of the true DNS data. The proposed PRU reduces the kinetic energy gap with the true DNS by 30% and 67% compared to TM and rTM, respectively. It confirms that PRU can better preserve underlying physical characteristics of turbulent flows (**E2**).

### 4.3.2 DNS reconstruction using PGSR-PRU

We implement the DNS reconstruction using PGSR-PRU and SRGAN-TM using the LES data for every five time steps (**E3**). As shown in Table 2 and Fig. 6, PGSR-PRU produces better performance than SRGAN-TM. Particularly in the first 15 time steps, it is more clear to see PGSRN-PRU can obtain lower RMSE and higher SSIM values. Fig. 6 also shows that the reconstruction performance gets degraded over time because the LES data have a significant difference with the training period. More interestingly, we notice that PGSR-PRU even outperforms PGSR (LES) and rPGSR (LES). This is because LES data often miss many important physical components compared to the true DNS, which makes SR models difficult to recover flow data directly

Table 3: Evaluation of SR models in terms of the reconstruction RMSE and SSIM on $(u, v, w)$ channels using LES data. The performance is measured on the testing data of the first 5 time steps.

| Method | RMSE | SSIM |
|---|---|---|
| RCAN | (0.061,0.061,0.075) | (0.891,0.891,0.863) |
| DCS/MS | (0.085,0.086,0.115) | (0.896,0.897,0.845) |
| SRGAN | (0.065,0.062,0.067) | (0.901,0.913,0.875) |
| PGSR-D | (0.057,0.052,0.053) | (0.914,0.923,0.900) |
| PGSR | (0.053,0.050,0.051) | (0.924,0.935,0.911) |

from LES data. We also show three sets of examples of reconstructed slides of flow data in Fig. 7. It is clear to observe that PGSR-PRU can better capture the detailed flow patterns compared to other methods as it incorporates the underlying Navier-Stokes equation through PRU.

### 4.3.3 DNS reconstruction using PGSR

As shown in Table 3, PGSR achieves better performance than other baselines in terms of both RMSE and SSIM. In particular, we can observe the improvement from SRGAN to PGSR-D and from PGSR-D to PGSR. This confirms the effectiveness of the degradation process and the physical constraints used in PGSR (**E4**).

## 5 CONCLUSION

We propose a physics-guided neural network framework for predicting high-resolution flow data at high temporal frequency. The PRU structure leverages the physical knowledge embodied in the Navier-Stokes equation to capture the flow dynamics over time while the PGSR model incorporates additional physical constraints to improve the reconstruction from the LES data. We have demonstrated the superiority of PRU in predicting future DNS data from historical DNS data. We also show that PGSR-PRU can effectively reconstruct DNS from sparse LES series.

More importantly, the proposed method is generally applicable to many scientific problems with similar properties, e.g., complex temporal dynamics, and the availability of low-resolution simulations with reduced accuracy. The PRU structure can also be used as a building block to enhance existing deep learning models for modeling of complex dynamics with the guidance of known governing PDEs.

## 6 ACKNOWLEDGEMENTS

The material presented in this paper is based upon work supported by the National Science Foundation (NSF) through Grant 2028001 and Grant OAC-2203581, the Defense Advanced Research Projects Agency (DARPA) under contract

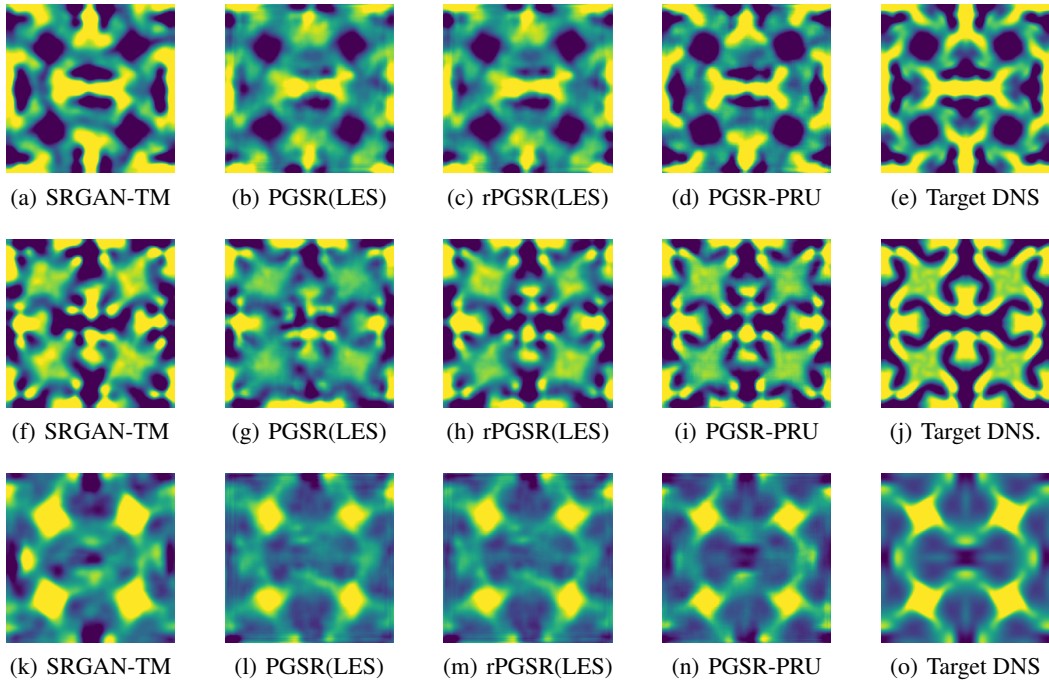

(a) SRGAN-TM    (b) PGSR(LES)    (c) rPGSR(LES)    (d) PGSR-PRU    (e) Target DNS

(f) SRGAN-TM    (g) PGSR(LES)    (h) rPGSR(LES)    (i) PGSR-PRU    (j) Target DNS.

(k) SRGAN-TM    (l) PGSR(LES)    (m) rPGSR(LES)    (n) PGSR-PRU    (o) Target DNS

Figure 7: Three example slides of reconstructed $w$ channel (in three rows) along the $z$ dimension with the LES interval of 5 time steps.

number FA8750-18-C-0089, and the Air Force Office of Scientific Research (AFOSR) under contract number FA9550-22-1-0019. Any opinions, findings, and conclusions or recommendations expressed in this paper are those of the authors and do not necessarily reflect the views of AFOSR, DARPA, or NSF. Computational resources are provided, in part, by the the University of Pittsburgh Center for research Computing (CRC).

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
