# OpenReview forum: "Physics Guided Neural Networks for Spatio-temporal Super-resolution of Turbulent Flows"
_auai.org/UAI/2022/Conference — UAI 2022 Poster_

### Official Review · Reviewer_vk7a · 2022-04-09

**Q2(1) Originality/Novelty:** 3
**Q2(2) Significance/Impact:** 3
**Q2(3) Correctness/Technical Quality:** 3
**Q2(6) Clarity Of Writing:** 3
**Q6 Overall Score:** 6
**Q8 Confidence In Your Score:** 3

**Q1 Summary And Contributions:**

In this work the authors propose a model for simulating turbulent flows that deals with the computational complexity of Direct Numerical simulation (DNS) method. The method takes  Low-resolution large eddy simulation (LES) and upscales the temporal and spatial resolution to decrease the gap in performance between LES and DES, without paying the cost of running DES. The performance of the method was evaluated in the Taylor-Green Vortex system.

**Q2 Assessment Of The Paper:**

More detailed information regarding each of these aspects is given below:

**Q2(4) Quality Of Experiments (Optional):**

2: Fair: The experimental evaluation is weak: important baselines are missing, or the results do not adequately support the main claims.

**Q2(5) Reproducibility:**

3: Good: Key resources (e.g., proofs, code, data) are available and key details (e.g., proofs, experimental setup) are sufficiently well-described for competent researchers to confidently reproduce the main results.

**Q3 Main Strengths:**

The main strength of this method is the effective integration of methods from Physics and Machine Learning (ML). The approach of using ML to estimate corrective terms in high cost computational simulation methods is taking momentum in the field and this is another method that is well positioned to demonstrate the value of this direction.
Particularly the special purpose models designed to approximate the computations in the Runge Kutta method and upscale the resolution of the LES method are the main contribution of the work.

**Q4 Main Weakness:**

The main weakness I see is in the results. We can see only minor improvement over SRGAN-TM . The improvement is generally in the few initial steps and then it diminishes.
So the question arises if there is sufficient evaluation in the paper to understand the cause of such performance of the models.
One aspect that I didn't fully understand and may be important is how the models are trained. From what I gathered the PRU model is training on DNS data over time. The PGSR is trained on mapping LES to DNS data. However, PRU does not seem to be exposed to errors from the PGSR method during training, so it does not get the opportunity to learn how to correct for those. If the model is trained (or fine-tuned) end-to-end such that PRU gets PGSR data while the loss is computed on DNS data, it may produce help with the overall performance.
The other key aspect is the baseline methods. Comparison with "Fourier Neural Operator for Parametric Partial Differential Equations" would be important.

**Q5 Detailed Comments To The Authors:**

There a few grammatical errors in the paper.

For example: "E3: How is the reconstruction performance combining PRU and PGSR using sparse low-resolution LES data?"

**Q7 Justification For Your Score:**

This work have solid ideas. The main drawback is stronger evaluation with existing SoA methods and relatively small improvement in performance with the currently evaluated methods.

**Q9 Complying With Reviewing Instructions:**

1: Yes.

---

### Official Review · Reviewer_E23d · 2022-04-14

**Q2(1) Originality/Novelty:** 2
**Q2(2) Significance/Impact:** 2
**Q2(3) Correctness/Technical Quality:** 2
**Q2(6) Clarity Of Writing:** 2
**Q6 Overall Score:** 4
**Q8 Confidence In Your Score:** 3

**Q1 Summary And Contributions:**

The authors developed a physics-guided neural network for recovering direct numerical simulation of turbulent flows from low resolution large eddy simulation. The approach consists of a partial differential equation recurrent unit for capturing underlying spatio-temporal processes and incorporating additional physical constraints. They demonstrated the effectiveness of both components in reconstructing the Taylor-Green Vortex using sparse LES data.

**Q2 Assessment Of The Paper:**

More detailed information regarding each of these aspects is given below:

**Q2(4) Quality Of Experiments (Optional):**

2: Fair: The experimental evaluation is weak: important baselines are missing, or the results do not adequately support the main claims.

**Q2(5) Reproducibility:**

2: Fair: Key resources (e.g., proofs, code, data) are unavailable but key details (e.g., proof sketches, experimental setup) are sufficiently well-described for an expert to confidently reproduce the main results.

**Q3 Main Strengths:**

Application deep learning to solving partial differential equations in engineering problems.

**Q4 Main Weakness:**

Application deep learning to solving partial differential equations is not new. Overall, I do not see stability and accuracy related analysis of the proposed approach, and I do not see the generalizability either.

**Q5 Detailed Comments To The Authors:**

- What are the theoretical guarantees and  limitations of the proposed work in computation time, generalizability, and robustness to noise?
- Can we start from the derivation from turbulent flows dynamics instead of intuitive description?

**Q7 Justification For Your Score:**

Application deep learning to solving partial differential equations is interesting. But the work is premature.

**Q9 Complying With Reviewing Instructions:**

1: Yes.

---

### Official Review · Reviewer_xoLg · 2022-04-18

**Q2(1) Originality/Novelty:** 2
**Q2(2) Significance/Impact:** 2
**Q2(3) Correctness/Technical Quality:** 3
**Q2(6) Clarity Of Writing:** 3
**Q6 Overall Score:** 6
**Q8 Confidence In Your Score:** 4

**Q1 Summary And Contributions:**

This paper presents a twofold technique for super-resolution of turbulent flows. The spatial super-resolution and the temporal super-resolution schemes are proposed. Provided comparison with other methods on a synthetic dataset indicate that the proposed methodology is on par or better in the considered scenarios.

**Q2 Assessment Of The Paper:**

More detailed information regarding each of these aspects is given below:

**Q2(4) Quality Of Experiments (Optional):**

2: Fair: The experimental evaluation is weak: important baselines are missing, or the results do not adequately support the main claims.

**Q2(5) Reproducibility:**

3: Good: Key resources (e.g., proofs, code, data) are available and key details (e.g., proofs, experimental setup) are sufficiently well-described for competent researchers to confidently reproduce the main results.

**Q3 Main Strengths:**

This paper presents a twofold technique for super-resolution of turbulent flows. The spatial super-resolution is achieved via SRGAN with augmented physics-guided loss function that enforces that properties of the flow are satisfied and a cycle-consistency-like loss. The temporal super-resolution is achieved by disentangling time and space derivatives in the ODE, replacing the space derivatives with learned convolutional discrete operators and the time derivatives with a partially learned Runge-Kutta-like solver.

Overall the paper is sound and the results are presented in an appropriate way. The proposed methodology for temporal super-resolution is an intriguing fusion of numerical methods and deep learning. It proves efficient over pure deep learning. The methodology for spatial super-resolution is as well promising, although could be further improved. The data section is well-detailed and code is provided, aiding reproducible research.

**Q4 Main Weakness:**

The paper is lacking comparison to principally different approaches, either simple ones or current state-of-the-art. Some details in the experiment section suggest the presence of overfitting that is not examined.


**Q5 Detailed Comments To The Authors:**

⁃ Stability analysis (Section 3.1.3) is not used.

 ⁃ «A degradation process» in Section 3.2 is not described.

 ⁃ The experimental part lacks details of experimental environment and the network architectures used.

 ⁃ The comparison to simple baselines (e.g. persistency baseline for temporal super-resolution and bicubic interpolation for spatial super-resolution) and to the state-of-the-art would also benefit this paper.

 ⁃ Ablation study is present for spatial super-resolution only, but the choice of RK4 scheme and SD layer is not motivated by an experiment.

 ⁃ The dataset is small and there is a clear sign of overfitting to train LES data in the results of Section 4.3.2. Using more data or augmenting existing data might help further improve the results.

 ⁃ The methodology for spatial super-resolution is limited to only a few physical properties, and can be further improved by incorporating e.g. the discussed kinetic energy into loss function.

 ⁃ The tables are not readable, I suggest highlighting the best results.

**Q7 Justification For Your Score:**

The paper proposes an interesting methodology that is worth sharing after careful revision. The revision is necessary to reduce possible overfitting effect and compare the results to simple baselines.

===

Comments were generally addressed except for overfitting. There is no a word about it except promises to show more results on other folds (?). The paper is about a new method, which looks interesting. However, the experimental section seems weak to me. Overall, I am OK to increase the score.

**Q9 Complying With Reviewing Instructions:**

1: Yes.

---

### Decision · Program_Chairs · 2022-05-15

**Decision:**

Accept (Poster)

**Comment:**

Meta Review: This paper on spatial and temporal super-resolution of turbulent flows. The novelty is acknowledged by Reviewer 1 & 2, while Reviewer 3 pointed out that the paper lacks theoretical analysis and formulation. Reviewer 1&2 also raise conern on the risk of overfiiting, and suggest a clear description of experiment settings and full comparison with more recent baselines. In all, the meta-reviewer considers the pros to outweigh the cons, and recommends acceptance. The authors need to incorporate the comments when preparing the final version